# The Association between Levels of Physical Activity and Lifestyle, Life Expectancy, and Quality of Life in Patients with Alzheimer’s Disease

**DOI:** 10.3390/jcm12237327

**Published:** 2023-11-26

**Authors:** Shima Gholamalishahi, Seyed Ali Hosseini, Evaristo Ettorre, Alice Mannocci, Chidiebere Emmanuel Okechukwu, Mohamad Javad Keshavarz, Giuseppe La Torre

**Affiliations:** 1Department of Public Health and Infectious Diseases, Sapienza University of Rome, 00161 Rome, Italy; shima.gholamalishahi@uniroma1.it (S.G.); chidiebere.okechukwu@uniroma1.it (C.E.O.); 2Department of Sport Physiology, Islamic Azad University, Shiraz 7198774731, Iran; alihoseini575757@gmail.com; 3Department of Cardiovascular, Respiratory, Nephrological, Anesthetic and Geriatric Sciences, Sapienza University of Rome, 00161 Rome, Italy; evaristo.ettorre@uniroma1.it; 4Faculty of Economics, University Mercatorum, 00186 Rome, Italy; alice.mannocci@unimercatorum.it; 5Department of Sports Management, Islamic Azad University of Yasuj, Yasuj 7591493686, Iran; keshavarzjavad@rocketmail.com

**Keywords:** Alzheimer’s disease, physical activity, lifestyle, life expectancy, quality of life

## Abstract

Background: Engaging in physical activity could help improve the quality of life in patients with Alzheimer’s disease. The objective of this study was to determine the association between the levels of physical activity and lifestyle, life expectancy, and quality of life in patients with Alzheimer’s disease in Iran and Italy. Methods: A total of 165 participants from Iran and Italy were enrolled in this cross-sectional study. In Iran, 85 patients participated in the study. In Italy, we enrolled 80 patients at the Sapienza University teaching hospital, Policlinico Umberto 1 in Rome. The inclusion criteria in this study include patients over 60 years old, non-smokers, and non-users of antidepressants and hypnotics. Results: The results of Tukey’s post hoc test of the study conducted in Iran showed that the lifestyle of patients with moderate (*p* = 0.001) and low (*p* = 0.009) physical activity levels was significantly better than inactive patients. Life expectancy in patients with moderate physical activity levels was significantly higher than inactive patients (*p* = 0.011). The quality of life was significantly better in patients with moderate (*p* = 0.001) and low (*p* = 0.002) physical activity levels than inactive patients. On the other hand, the findings of Tukey’s post hoc test of the study in Italy showed that the quality of life in patients with low (*p* = 0.001) and moderate physical activity levels (*p* = 0.01) was significantly higher than inactive patients. Conclusions: A low to moderate level of physical activity could be associated with an improved lifestyle, life expectancy, and quality of life in patients with Alzheimer’s disease compared to inactivity.

## 1. Introduction

Alzheimer’s disease (AD) is a neurodegenerative disorder that predominantly affects memory, thinking, and behavior and is a common type of dementia [1]. The early symptoms of AD often include difficulty remembering recent events, confusion, changes in mood and behavior, and challenges with language and problem solving. As the disease progresses, individuals may experience severe memory loss, disorientation, difficulty speaking and swallowing, and changes in personality [2]. Due to the aging of the global population, there is an increase in dementia, which is becoming one of the biggest problems in public health, clinical, and social challenges. It is estimated that around 55 million people were living with dementia in 2019, and the number may increase to 139 million in 2050 [3]. Currently, AD affects millions of people around the world and adversely affects their quality of life (QOL) [4,5]. Since there is no effective treatment for AD, prevention through lifestyle interventions is critical [6]. Health-related quality of life (HRQOL) influences vast aspects of health like physical and mental health, autonomy, social interaction, and the relationship between a subject and their environment [7]. Most patients with AD experience neuropsychiatric symptoms and cognitive impairment during the course of their illnesses, such as apathy, agitation, and psychosis, which negatively affect their HRQOL [8]. Physical activity (PA) has been linked to improved cognitive function and memory in people with Alzheimer’s disease. It can help slow down the decline in thinking skills and delay the onset of symptoms [9]. Numerous systematic reviews and meta-analyses have indicated that engaging in PA may play a crucial role in reducing or postponing the emergence of various modifiable risk factors associated with cognitive decline such as obesity, diabetes, and hypertension [3,10]. A study by Ya-jing et al. demonstrated that short-term (2–5 months) PA interventions were more beneficial in improving cognitive function, neuropsychiatric symptoms, and QOL in patients with AD [11]. Italy and Iran had an increasing aging population [12]. Italy already has the highest median age (46.7 years) and the highest share of elderly people (22.8% of the country’s population is over 65) in Europe. Taking current aging trends into account, the report estimates that over 2.2 million people in Italy will be living with dementia in 2050, almost double the current number [13]. In Iran, like many other countries, the aging demographic is expanding, and it is expected that 8–10% of the elderly people in Iran will be affected by this disease over the next 2–3 decades. People over the age of 60 suffering from AD spend around 11.2% of their lives with a disability [14]. 

PA antioxidant and anti-inflammatory effects have shown a positive impact on aging and neurodegeneration and represent a possible treatment option for cognitive decline [15]. On the other hand, physical inactivity has been recognized as a risk factor for increased memory loss and the development of AD [16]. Given the rising aging population in Italy and Iran, we examine two distinct patient groups on different continents, each navigating unique circumstances. This study aims to explore the correlation between levels of physical activity and lifestyle factors and their impact on life expectancy and quality of life among patients with AD in both Iran and Italy. 

## 2. Materials and Methods

### 2.1. Study Design

Because of the researchers’ expertise in the cultures of both nations, this cross-sectional study was conducted in two specific locations in Iran and Italy. The researchers reasoned that if the outcomes of the two studies were similar, the findings may apply to countries in the rest of the world. The study was conducted from July to October 2020 in Iran and from July to December 2021 in Italy. Patients were recruited from the AD Association in Shiraz Province, Iran, and the Sapienza University of Rome Teaching Hospital, Policlinico Umberto 1, Rome, Italy. 

### 2.2. Ethics 

The protocol was designed following the recommendations of the Consolidated Standards of Reporting Trials statements [17]. This study was conducted following the Declaration of Helsinki and was approved by the Research Ethics Committee of the University of Marvdasht Branch IR.IAU.M.REC.1399.049 and the University of Sapienza Prot. 0982/2021. Each person likely to take part in this research was informed, and a summary document on the main characteristics of the study was given during the inclusion visit. The written informed consent statement was given for immediate signature or after a reflection period. In case of doubt about the patient’s ability to give consent, caregivers and the healthcare team were asked to help measure the expression of the person’s autonomy and presumed wishes. 

### 2.3. Participants and Inclusion Criteria 

A total of 165 participants from both countries were enrolled in this study. In Iran, 85 patients participated in the study. In Italy, we enrolled 80 patients at the Sapienza University teaching hospital, Policlinico Umberto 1 in Rome. Participants for this cross-sectional study were recruited based on specific inclusion and exclusion criteria. Inclusion criteria in the present study include individuals diagnosed with AD and classified as level 1 or 2 of AD who were over 60 years old, non-smokers, and non-users of antidepressants and hypnotics. Physical activity, lifestyle, life expectancy, and quality of life questionnaires were completed by the participants under the supervision of their caregivers, who live with them or spend considerable time with them. In Iran, the mean age of the participants was 72.89 years (SD = 9.90). Thirty-seven percent were males and sixty-two percent were females. In Italy, the mean age of the patients was 80.44 years (SD = 6.73). A total of 36.7 percent of the population were males and 46.7 percent of the population were females. 

### 2.4. Physical Activity Assessment

In our study, we used the abbreviated form of the International Physical Activity Questionnaire (IPAQ) to evaluate the participants’ levels of physical activity [18]. The seven self-reported questions on the questionnaire estimate physical activity levels from the previous week [18]. According to this questionnaire, the metabolic equivalent of walking is 3.3 MET, a moderate walk is 4 MET, and an intense walk is 8 MET. The walking (day × minute × MET), moderate (day × minute × MET), and severe (day × minute × MET) physical activity per week were combined to obtain the total amount of physical activity per week. Walking, moderate exercise, and intensive activity were found to have different energy expenditures based on the formula (MET-min/week). If the sum of MET-min/week in walking, moderate, and intense activity for seven days was at least 3000 or more, between 600 and 3000, and lower than 600; it was categorized as high physical activity, moderate physical activity, and low physical activity, respectively. Individuals who did not report any physical activity were categorized as inactive. 

### 2.5. Lifestyle Assessment 

Globally, a person’s lifestyle is one of the main factors that determines their general health. Participants’ lifestyles were assessed using a Miller and Smith lifestyle questionnaire. The Miller–Smith Lifestyle Assessment Inventory consists of twenty items with a 5-point Likert-type scale asking respondents how often the connected items apply to them [19]. The patients’ general health, weight, food, nicotine use, affection, exercise, sleep patterns, spirituality, and social connections were all assessed using it. Responses range from 1 (always) to 5 (never). The total score ranges from 20 to 100. The statement scores for each component were tallied, divided by the total number of items, and expressed as percentage values. Therefore, each component’s mean score was determined. Higher rankings signify an unhealthy and bad way of living. A total score of less than 50% indicated a great or healthy way of life, a score of 50% to 70% indicated very good/moderate, a score of 70% to 95% indicated good/mild, and a score of greater than 95% indicated unhealthy or poor. In 1988, Miller and Smith reported that the reliability was α = 0.85. Research data were analyzed using structural equation models (*p* < 0.001) using the analysis of moment structures (version 22.0, Chicago, IL, USA: IBM SPSS).

### 2.6. Quality of Life Assessment

We used a brief version of the World Health Organization quality of life scale (WHOQOL-BREF) in this study [20]. The WHOQOL-BREF questionnaire contains 2 items from the overall QOL and general health and 24 items of satisfaction that are divided into 4 domains: physical health with 7 items: questions 3, 4, 10, 15, 16, 17, and 18 (DOM1), psychological health with 6 items: questions 5, 6, 7, 11, 19, and 26 (DOM2), social relationships with 3 items, questions 20, 21, and 22 (DOM3), and environmental health with 8 items: questions 8, 9, 12, 13, 14, 23, 24, and 25 (DOM4) [20]. For scoring, the scores of each item are in the range of 1 to 5, respectively: not at all, low, medium, high, and completely; or very dissatisfied, dissatisfied, relatively dissatisfied, satisfied, and completely satisfied. 

### 2.7. Life Expectancy Assessment

Schneider’s Life Expectancy Questionnaire is a 12-item assessment tool designed for evaluating an individual’s life expectancy [21]. Four items are used to measure agency thinking, four are used to measure strategic thinking, and four are considered deviant statements in the questionnaire. Agency and strategy are its two subscales. This questionnaire evaluates two aspects of life expectancy: (1) the drive to accomplish life’s objectives and (2) the individual strategy for accomplishing those objectives. The Likert scale has five points: 1 for completely disagree and 5 for totally agree. This is the basis for the scoring. The overall hope score is the product of the strategy and factor subscale scores. Over a ten-week period, dependability is measured at 80% and Cronbach’s alpha ranges from 74% to 84%. The reliability and accuracy of the questionnaire as a means of assessing life expectancy have been validated by numerous researchers [22]. The test’s internal consistency ranges from 0.74 to 0.84, and its test–retest validity is 0.80. These numbers would likely increase if the test was administered over an extended period.

### 2.8. Statistical Analysis

The Kolmogorov–Smirnov test was used to assess the normal distribution of findings. Due to the abnormal distribution of physical activity findings, Kruskal–Wallis and Mann–Whitney U tests were used. Also, due to the natural distribution of lifestyle findings, life expectancy, and quality of life along with the physical health, psychological, social relations, and environmental domains, a one-way analysis of variance test (ANOVA) was used along with Tukey’s post hoc test. The Pearson correlation coefficient test was used to examine the correlation between physical activity and lifestyle, life expectancy, and quality of life (*p* ≤ 0.05). Multiple linear regression analyses with cognition, functional capacity, and QOL as the dependent variables and physical activity, lifestyle, and life expectancy as the independent variables were used.

## 3. Results

### 3.1. The Association between the Levels of Physical Activity and Lifestyle, Life Expectancy, and Quality of Life in Patients with Alzheimer’s Disease in Italy

The sample consists of 165 subjects (80 Italian; 85 Iranian); 58.2% are female. The educational level reported includes 17.8% graduates of the first and second level, 10% high school graduates, 25.5% primary school level, and 42% illiterate. The age of all subjects ranged from 56 to 95 years (mean 75.9 years SD = 9.40 yrs).

The characteristics of the participants satisfied by gender are summarized in Table 1.

The results of the one-way ANOVA (Table 2) showed a significant difference in QOL along with domains of physical health, psychological functioning, social relationships, and environmental factors in inactive, low physical activity, and moderate physical activity AD patients (*p* ≤ 0.05). The findings of Tukey’s post hoc test (Table 2) showed that QOL in patients with low (*p* = 0.001) and moderate physical activity levels (*p* = 0.01) was significantly higher than inactive patients. The physical health domain (in quality of life) in patients with low physical activity levels (*p* = 0.01) was significantly higher than inactive patients and patients with moderate physical activity levels. The psychological domain in patients with low physical activity was significantly higher than inactive patients (*p* = 0.001) and patients with moderate physical activity levels (*p* = 0.01). Furthermore, the social relation domain in patients with moderate physical activity levels (*p* = 0.001) was significantly higher than patients with low physical activity levels (*p* = 0.01) and inactive patients. The environmental domain variables of patients with moderate physical activity levels (*p* = 0.02) were significantly higher than in patients with low physical activity and inactive patients (Table 2). The results of the Pearson coefficient showed a significant direct correlation between physical activity level and QOL (*p* = 0.002) in patients with AD (Table 3). The corresponding β-coefficients from multivariable linear regression models, adjusted for lifestyle, life expectancy, and physical activity, are presented in Table 4. There was no significant association between QOL as a dependent variable and lifestyle, life expectancy, and physical activity as independent variables.

### 3.2. The Association between the Levels of Physical Activity and Lifestyle, Life Expectancy, and Quality of Life in Patients with Alzheimer’s Disease in Iran

The physical activity attributes of the subjects are reported in Table 1. Also reported are the levels of lifestyle, life expectancy, physical activity, and quality of life, along with the domains of physical health, psychological health, social relations, and the environment (Table 2). The results of the one-way ANOVA (Table 2) showed a significant difference in lifestyle, life expectancy, QOL, and domains of physical health, psychological health, social relationship, and environmental factors in inactive, low physical activity, and moderate physical activity AD patients (*p* ≤ 0.05). The results of Tukey’s post hoc test (Table 2) showed that the lifestyle of patients with moderate (*p* = 0.001) and low (*p* = 0.009) physical activity levels was significantly better than inactive patients. Life expectancy in patients with moderate physical activity was significantly higher than inactive patients (*p* = 0.011). Quality of life was significantly better in patients with moderate (*p* = 0.001) and low (*p* = 0.002) physical activity levels than inactive patients (Table 2). The physical health domain (in QoL) in patients with moderate physical activity levels was significantly higher than inactive patients (*p* = 0.001) and low physical activity (*p* = 0.008) levels, and the psychological domain in patients with moderate physical activity levels was significantly higher than inactive patients (*p* = 0.001) and low physical activity (*p* = 0.03). Also, in patients with low physical activity levels, it was significantly higher than inactive patients (*p* = 0.009). The social relation domain in patients with moderate physical activity (*p* = 0.001) and low physical activity levels (*p* = 0.008) was significantly higher than inactive patients. Also, the environmental domain in patients with moderate physical activity (*p* = 0.007) and low physical activity (*p* = 0.001) levels was significantly higher than inactive patients. The results of the Kruskal–Wallis test showed that there was a significant difference in the level of physical activity in AD patients (*p* = 0.001). The results of the Pearson correlation coefficient test showed a significant positive relationship between physical activity and lifestyle improvement (*p* = 0.002), life expectancy (*p* = 0.001), and quality of life (*p* = 0.001) in AD patients (Table 3). To determine the multiple linear regression equation, we considered QOL as a dependent variable and physical activity, lifestyle, and life expectancy as independent variables (Table 4). The results of this test demonstrated that among women, lifestyle (*p* = 0.00) (β = −0.78) and life expectancy (*p* = 0.03) (β = 0.71) had a significant effect on the QOL in patients with AD. Lifestyle in men (*p* = 0.014) (β = −0.611) also had a significant effect on their QOL, but in women, it is better than in men (Table 4). Furthermore, physical activity has a significant effect on QOL (*p* = 0.022) (Table 4).

## 4. Discussion

The optimal level of physical activity/dose of exercise required to achieve improvements in functional and prognostic parameters among older adults remains an issue of global debate among healthcare practitioners [23]. According to the results of our study, we found a significant association between QOL and lifestyle (*p* = 0.001), life expectancy (*p* = 0.003), and physical activity (*p* = 0.010) in patients in Iran and Italy (Table 5). The results of the study we conducted in Iran showed that compared to inactive patients, AD patients with low to moderate levels of physical activity had significantly better lifestyles, quality of life, social relationships, psychological functioning, and living conditions. This is consistent with a study conducted by Iranian researchers in Shiraz, which found that cognitive impairment was more common in elderly people with a lack of physical activity [24]. Most of the longitudinal epidemiological research has conclusively demonstrated dose–response relationships between physical activity and the risk of cognitive decline, indicating that physical activity may delay the development of AD as well as the risk of cognitive decline and mortality [10]. Even in extended follow-ups for all-cause dementia and AD, physical activity was linked to a lower risk of dementia from all causes, Alzheimer’s disease, and vascular dementia [25]. Primary prevention could result in a meaningful decline in AD cases in Italy [26]. The results of our study in Italy showed that quality of life, social relationships, psychological health, and living environment were significantly and positively associated with low to moderate physical activity in patients with AD. This is consistent with the findings of a nationwide cohort study in Korea that included 62,286 participants and found that an increased physical activity level, including a low amount of light-intensity activity, was associated with a lower risk of developing AD [27]. AD was significantly and inversely related to the quality of life, and physical activity and was directly related to age, chronic diseases, and body mass index [7]. In addition, the relationship between quality of life and AD was affected by gender; in men, the quality of life of patients is affected more than women [7]. The effects of AD depend on factors such as gender, age, level of physical activity, and lifestyle. Long-term and regular physical activity is a method of preventing neuronal damage and memory loss [28]. In this regard, researchers have shown that exercise, regardless of its intensity as a non-pharmacological intervention, can have beneficial effects on quality of life in people with AD [29]. Also, combined exercise and resistance training improved social relationships and cognitive function in patients with AD [30]. One of the modifiable risk factors for cognitive impairment that can be prevented is sedentary behavior. Physical activity can be incorporated into programs relating to active aging because of its cardio-protective effects, which may promote brain activity and cognitive functioning in old adults with or without dementia [31]. Furthermore, regular physical activity was associated with an increase in life expectancy [32]. A recent study found that among older people with AD, physical activity interventions improved overall cognitive functioning [33]. The researchers discovered that weight training, moderate-intensity exercise, and exercise programs lasting more than 24 weeks all improved cognitive control, as did low-frequency therapies given once or twice a week [33]. Furthermore, the impact sizes of physical activity interventions on cognitive performance were comparable regardless of the duration of the exercise sessions. Individuals between the ages of 76 and 90 showed a greater favorable result for cognitive performance following physical exercise interventions compared to younger participants [33]. The outcome of their study demonstrates that designing and personalizing physical activity interventions will be effective in enhancing executive function and improving the quality of life for AD patients.

Lastly, the association between physical activity levels, lifestyle, life expectancy, and QOL in patients with AD is an important area of research. While there are few recent studies that have evaluated this topic, our study explored the significance of physical activity levels in individuals with AD in relation to their lifestyle and well-being. While physical activity may not directly impact life expectancy among patients with AD, it can have indirect benefits by improving overall health and promoting well-being. Tailored physical activity interventions may help manage comorbidities, such as cardiovascular conditions or diabetes, which can influence life expectancy among patients with AD. Physical activity can enhance functional capacity and independence, thus improving the overall QOL for people living with AD [34,35]. Individualized physical activity interventions tailored to the needs and capabilities of patients with AD can enhance mood, improve independence, reduce depressive symptoms, anxiety, and loneliness, and improve sleep patterns, leading to a better overall sense of well-being. It may also positively improve social interactions, cognitive function, and the ability to perform activities of daily living among older adults [34]. The outcome of our research suggests a positive association between physical activity, lifestyle, life expectancy, and QOL in patients with AD. It is important to note that AD is complicated, and individual responses to PA may vary with regard to cardiorespiratory fitness and clinical status [35]. 

## 5. Conclusions

The results of the study in Iran demonstrated that lifestyle, quality of life, social relationships, psychological functioning, and life expectancy were significantly higher in AD patients with low to moderate physical activity levels compared to inactive patients. The findings of the study conducted in Italy demonstrated that quality of life, social relationships, and psychological health in patients with AD were significantly associated with low to moderate physical activity. Having a low to moderate physical activity level could be associated with an improved lifestyle, life expectancy, and quality of life in patients with AD compared to inactivity. Therefore, progressive low- to moderate-intensity physical activity can be recommended to patients with AD as a health promotion and prevention strategy after careful consideration of the individual’s health status, training load/baseline fitness, goals, and preferences.

## 6. Strengths and Limitations

The strength of this study is that it shows that an active lifestyle could improve longevity and QOL in patients with AD. The inadequate number of patients who participated in this study was a limitation of this study, and this was due to the COVID-19 pandemic. Therefore, larger cohort and cross-sectional studies are recommended in the future. Also, looking at the pathological and physiological changes and adaptations to physical activity, it appears that the absence of adequate demographic and clinical characteristics of the patients in terms of clinical diagnosis, level of cognitive and behavioral impairment, comorbidities, and the use of anti-AD medications and their relationships with physical activity levels is the major limitation of the present study. Therefore, utilizing sufficient demographics and clinical data of patients is suggested in future studies.

## Figures and Tables

**Table 1 jcm-12-07327-t001:** Physical activity attributes of the subjects.

Physical Activity Group	Italian Sample (N = 80)			Iranian Sample (N = 85)		
	Age (Mean ± SD)	Gender		Age (Mean ± SD)	Gender	
		Female N (%)	Male N (%)		Female N (%)	Male N (%)
Inactive	76.63 ± 1.17	22 (51.2)	24 (64.8)	68.25 ± 9.21	14 (87.5)	2 (12.5)
Low Physical Activity	79.37 ± 1.84	10 (23.2)	6 (16.2)	74.68 ± 9.77	26 (74.3)	9 (25.7)
Moderate Physical Activity	77.50 ± 1.73	11 (25.5)	7 (18)	73.23 ± 9.92	13 (38.2)	21 (61.8)
**Total**	**79.10 ± 7.67**	**43 (53.7)**	**37 (46.2)**	**72.89 ± 9.90**	**53 (62.4)**	**32 (37.6)**

**Table 2 jcm-12-07327-t002:** The results of the one-way ANOVA test with Tukey’s post hoc test compared the lifestyle, life expectancy, and quality of life among patients with Alzheimer’s disease.

	Variables	Inactive	Low Physical Activity	Moderate Physical Activity	*p*-Value ^f^	Male	Female	*p*-Value
		Mean (SD)	Mean (SD)	Mean (SD)		Mean (SD)	Mean (SD)	
Italian Sample	Lifestyle	54.84 (11.13) ^a,b^	44.50 (12.39) ^a^	39.77 (9.58) ^b^	**≤0.001**	51.00 (11.76)	48.00 (13.55)	0.297 ^d^
	Life expectancy	32.06 (8.80) ^a^	39.75 (8.37) ^a^	35.55 (11.15)	**0.018**	33.38 (9.84)	35.26 (9.56)	0.390 ^d^
	Quality of life	70.76 (14.81) ^a,b^	102.50 (7.19) ^a^	93.27 (12.15) ^b^	**≤0.001**	78.51 (17.63)	85.33 (19.44)	0.107 ^d^
	Physical health domain	19.78 (4.87) ^a,b^	27.31 (1.95) ^a^	25.44 (2.97) ^b^	**≤0.001**	21.84 (5.13)	23.19 (5.25)	0.251
	Psychological domain	16.19 (4.23) ^a,b^	25.56 (2.39) ^a^	22.27 (4.02) ^b^	**≤0.001**	18.24 (5.03)	20.47 (5.74)	0.072 ^d^
	Social relationship domain	7.71 (2.21) ^a,b^	11.81 (1.79) ^a^	10.77 (1.83) ^b^	**≤** **0.001**	8.86 (2.27)	9.53 (2.78)	0.274 ^d^
	Environmental domain	20.97 (4.21) ^a,b^	28.87 (4.01) ^a^	27.00 (3.53) ^b^	**≤** **0.001**	22.97 (4.93)	24.72 (5.52)	0.142 ^d^
Iranian Sample	Lifestyle	71.00 (13.44) ^a,b^	58.40 (16.13) ^a^	53.58 (11.18) ^b^	**≤0.001**	52.72 (12.59)	62.55 (15.29)	**0.003 ^d^**
	Life expectancy	35.50 (4.67) ^b^	37.62 (6.64) ^c^	41.41 (7.26) ^b,c^	**0.007**	40.61 (7.49)	37.61 (6.36)	**0.053 ^d^**
	Quality of life	54.50 (16.94) ^a,b^	72.20 (18.85) ^a^	79.78 (15.10) ^b^	**≤0.001**	81.90 (14.38)	65.86 (19.28)	**<0.001 ^e^**
	Physical health domain	14.31 (3.70) ^b^	17.71 (5.94) ^c^	21.48 (4.72) ^b,c^	**≤0.001**	21.38 (5.17)	16.99 (5.40)	**<0.001 ^d^**
	Psychological domain	10.06 (5.01) ^a,b^	14.60 (5.24) ^a,c^	17.70 (4.72) ^b,c^	**≤0.001**	17.97 (4.43)	13.19 (5.60)	**0.001 ^d^**
	Social relationship domain	5.43 (3.84) ^a,b^	8.54 (3.95) ^a^	9.66 (2.38) ^b^	**≤0.001**	10.37 (2.78)	7.22 (3.67)	**<0.001 ^e^**
	Environmental domain	19.75 (5.96) ^a,b^	25.65 (5.02) ^a^	24.59 (4.88) ^b^	**≤0.001**	26.41 (4.79)	22.74 (5.54)	**0.003 ^d^**

Bold: *p* ≤0.05. ^a^: low versus inactive groups with a *p*-value of *p* ≤ 0.016 (with Bonferroni’s test correction using adjusted alpha (α) = α/k, k is the number comparisons: 0.05/3 = 0.016). ^b^: moderate versus inactive groups with a *p*-value of *p* ≤ 0.016 (with Bonferroni’s test correction using adjusted alpha (α) = α/k, k is the number comparisons: 0.05/3 = 0.016). ^c^: moderate versus low groups with a *p*-value of *p* ≤ 0.016 (with Bonferroni’s test correction using adjusted alpha (α) = α/k, k is the number comparisons: 0.05/3 = 0.016). ^d^: t-student with equal variances assumed. ^e^: t-student with equal variances not assumed. ^f^: *p*-value of an ANOVA test.

**Table 3 jcm-12-07327-t003:** The results of the Pearson correlation coefficient test assess the relationship between physical activity, lifestyle, life expectancy, and quality of life in patients with Alzheimer’s disease.

		Italian Sample			Iranian Sample		
		Life Expectancy	Lifestyle	Quality of Life	Life Expectancy	Lifestyle	Quality of Life
Physical Activity (IPAQ score)	Pearson’s Coefficient	0.20	0.50	0.58	0.39	0.29	0.40
	*p*	0.07	**≤0.001**	**≤0.001**	**≤0.001**	**≤0.001**	**≤0.001**

Bold: *p* ≤ 0.05.

**Table 4 jcm-12-07327-t004:** Multiple linear regression model for quality of life.

	Covariates	All Patients				Females				Males			
		Full Model		Backward Elimination		Full Model		Backward Elimination		Full Model		Backward Elimination	
		Beta	* p *	Beta	* p *	Beta	* p *	Beta	* p *	Beta	* p *	Beta	* p *
Italian Sample	Lifestyle	−0.638	**≤0.001**	b		−0.564	**0.017**	−0.681	**≤0.001**	−0.691	**≤0.001**	b	
	Life expectancy	0.493	**0.007**			0.370	0.186	-	-	0.632	**0.008**		
	Physical activity	0.001	**0.046**			0.001	0.325	-	-	0.002	**0.014**		
	R^2^	0.300				0.282		0.248		0.610			
Iranian Sample	Lifestyle	−0.795	**≤0.001**	−0.858	**≤** ** 0.001 **	−0.789	**≤0.001**	−0.797	**≤0.001**	−0.611	** 0.014 **	−0.600	**≤0.002**
	Life expectancy	0.269	0.293	-	-	0.719	** 0.033 **	0.759	** 0.017 **	−0.031	0.939	-	-
	Physical activity	0.005	** 0.022 **	0.006	** 0.008 **	−0.001	0.707	-	-	0.004	0.207	-	-
	R^2^	0.584		0.579		0.631		0.630		0.316		0.273	

Bold: *p* < 0.05; b: no backward elimination steps.

**Table 5 jcm-12-07327-t005:** Linear regression model for quality of life considering the total sample (Iranian and Italian patients).

Covariates	QOL			
	Full Model (First Step)		Backward Elimination	
	Beta	* p *	Beta	* p *
Gender (male reference group)	−0.046	0.401	−0.046	0.402
Age	−0.047	0.420	−0.047	0.420
Nationality (Iranian reference group)	0.109	0.093	0.098	0.118
Lifestyle	−0.549	**<0.001**	**−0.585**	**<0.001 ***
Life expectancy	0.209	**0.001**	**0.175**	**0.003 ***
Physical activity	0.143	0.019	0.155	**0.010 ***
R^2^	0.516		0.513	

* Last step. QOL: quality of life. Bold: statistical significance.

## Data Availability

The data presented in this study are available on reasonable request from the corresponding author. The data are not publicly available due to privacy and ethical reasons.

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
