# Peer review of "The Association between Levels of Physical Activity and Lifestyle, Life Expectancy, and Quality of Life in Patients with Alzheimer’s Disease"

_jcm, 2023, doi:10.3390/jcm12237327_

Round 1

Reviewer 1 Report

Comments and Suggestions for Authors

Dear Authors

you are taking up a very important research topic regarding people struggling with Alzheimer's disease. The work requires many corrections, especially in the introduction, results and discussion.

I am sending all suggestions and comments recommending corrections within selected sections:

Introduction:

Please add research results confirming the beneficial effect of physical activity as a factor that may affect the quality of life of people with Alzheimer's disease. Please also raise issues related to the fact that physical activity undertaken together, in a group and with other patients or friends may play an important social and integrating factor for people with Alzheimer's disease.

Material and methods:

please add a graph in paragraph 2.2 presenting the course of the research experiment, taking into account the group size, number of participants, inclusion and exclusion criteria from the study

- did the subjects agree to participate in the research and how did they give this consent? Were the research voluntary and anonymous and the participants were informed about its purpose?

- Were the respondents instructed on how to complete the questionnaire in order to eliminate measurement errors?

- did any questionnaires contain errors? Were they rejected from the analysis and for what reason?

- were there people in the study group invited to participate in the study who refused to participate in the study?

- did the surveyed people complete the questionnaire on their own?

Results:

please present the most important results in the form of charts, which is more attractive to the reader

in table No. 2, please add columns broken down by gender, and also please calculate whether there was a correlation between the examined variables and the gender of the examined persons

discussion:

Please expand the discussion, especially the role of physical activity as a factor in the prevention of Alzheimer's disease, and a factor that can reduce its effects.

In the discussion, please also emphasize the social and integrative role of physical activity, which is especially important for sick people. Additionally, please add a paragraph showing that gender may influence the parameters you are examining. According to available literature, men are usually more physically active than women and can this have an impact on lifestyle and its quality later in life, especially in people suffering from Alzheimer's disease

Reviewer 2 Report

Comments and Suggestions for Authors

In this manuscript the authors analyzed the association between the levels of physical activity and lifestyle, life expectancy, and quality of life in patients with Alzheimer’s disease in Iran and Italy. The concept of the manuscript is interesting, and the results are presented in detail. However, there are certain things that need to be corrected:

1.     Introduction, lines 43 and 44: “Senescence is the most significant risk factor for AD, with most cases occurring in people over 65” ………If this sentence is already mentioned, the question for the authors is why the inclusion criterion for participation in the study was age over 60 years?

2.     Introduction, lines 56 and 57: the authors explained to some extent the reason why the population of Italy was included in the study. Accordingly, it is necessary to state the reason for the participation of the population originating from Iran, whether there is any similarity between the populations or a common characteristic?

3.     Study design: the authors did not specify what type of study it was, a cross-sectional study, a multicenter cohort study, etc…

4.     The record number of study approvals from the competent ethical committees is missing.

5.     Specify exclusion criteria!

6.     It would be desirable to add at least some socio-demographic characteristics in the part related to the description of the population; apart from age and gender distribution, there is no data on other demographic determinants

7.     Results, lines 156 and 157, delete the sentence: “The results of the Kolmogorov-Smirnova test showed that the data distribution was normal (P≥0.05)”

8.     Results, lines 180-212: adjust the spacing so that it matches the rest of the text!

9.     add the values of the odds ratios (OR) in the tables that list the results of the regression analysis

10.  It would be interesting to compare the data obtained in Italy with the data obtained from Iran, in order to make differences between determinants, predictors of the condition of patients with Alzheimer’s disease
